# Challenges in Diagnosing Pediatric Monteggia Fractures: Role of Clinical Experience and Imaging

**DOI:** 10.3390/medicina61081457

**Published:** 2025-08-13

**Authors:** Min Hyeok Choi, Min Hui Moon, Suk Woong Kang, Kyeong Baek Kim, Tae Young Ahn, Jung Yun Bae

**Affiliations:** 1Department of Preventive and Occupational and Environmental Medicine, School of Medicine, Pusan National University, Yangsan 50612, Republic of Korea; come2mh@gmail.com (M.H.C.); pnuyh3255@gmail.com (M.H.M.); 2Office of Public Healthcare Service, Pusan National University Yangsan Hospital, Yangsan 50612, Republic of Korea; 3Department of Orthopedic Surgery, Pusan National University Yangsan Hospital, Yangsan 50612, Republic of Korea; redmaniak@naver.com (S.W.K.); pnuyh.102@gmail.com (K.B.K.); 4Department of Orthopedic Surgery, School of Medicine, Pusan National University, Yangsan 50612, Republic of Korea; aa01066863594@gmail.com; 5Department of Orthopedic Surgery, Pusan National University Hospital, Busan 49241, Republic of Korea

**Keywords:** Monteggia fracture, pediatric trauma, diagnostic accuracy

## Abstract

*Background and Objectives*: Monteggia fracture-dislocations are rare but critical injuries in children. Accurate early diagnosis is essential to avoid long-term complications; however, such injuries are frequently missed. Therefore, this study aimed to assess the diagnostic accuracy of Monteggia fractures among physicians of varying specialties and experience levels and to identify factors influencing diagnostic performance. *Materials and Methods*: This retrospective study analyzed the radiographic interpretations of pediatric elbow and forearm injuries by six physician groups: orthopedic residents, general orthopedic surgeons, pediatric orthopedic surgeons, general radiologists, and subspecialized musculoskeletal radiologists. The final diagnosis established by pediatric radiology experts served as the reference standard. Influential variables, such as image quality, splint application, and age-related ossification, were evaluated. *Results*: In total, 120 patients were included, 40 (33.3%) of whom were diagnosed with Monteggia fractures and 80 (66.7%) with other fracture types. The diagnostic accuracy of Monteggia fractures varied significantly according to the physician’s experience. First-year residents and non-subspecialty radiologists had the highest rate of missed diagnoses. While other fracture types were occasionally influenced by technical factors, most missed Monteggia fracture cases stemmed from recognition failure. Subtle imaging features, plastic deformation of the ulna, and the omission of dedicated elbow views contributed to the misdiagnosis. Awareness and training improved performance, and a high index of suspicion was identified as crucial. Early follow-ups and standardized imaging protocols were identified as effective safeguards. *Conclusions*: Experience level, awareness, and imaging protocol quality were identified as being central to the accurate diagnosis of pediatric Monteggia fractures. Implementing educational strategies, promoting systematic imaging reviews, and reinforcing team-based approaches may reduce the rate of missed diagnoses.

## 1. Introduction

Monteggia fracture-dislocations are uncommon but clinically significant injuries in children, accounting for approximately 1–2% of all pediatric fractures [1,2]. This lesion, defined as an ulnar fracture or plastic deformation of the ulna, is accompanied by the dislocation of the radial head [3]. When recognized early and treated appropriately, outcomes are typically favorable, with the restoration of elbow stability and near-normal range of motion [4].

Management in children initially involves the urgent anatomical realignment of the ulna and a reduction in the radial head. This is most commonly achieved through closed reduction and casting as a first-line conservative approach. If successful, this method stabilizes the radiocapitellar joint and restores elbow function without requiring surgical intervention (Figure 1) [4]. However, in cases in which the alignment is unstable or cannot be maintained, surgical stabilization using techniques such as intramedullary nailing or plating is indicated [5,6]. Therefore, a prompt and accurate diagnosis is critical for determining the appropriate treatment pathway and avoiding long-term complications, such as persistent radial head dislocation, limited motion, and elbow deformity.

When diagnosis is delayed beyond the acute phase, the injury is considered a “neglected Monteggia fracture,” typically defined as untreated for more than two weeks. As illustrated in Figure 2, delayed treatment can result in chronic dislocation of the radial head, leading to progressive pain, valgus deformities, and a restricted range of motion [2,7]. Irreducibility due to soft tissue contracture or annular ligament entrapment may develop over time, and altered joint mechanics can result in degenerative changes and ulnar nerve dysfunction [2,8]. Surgical reconstruction (Figure 3), often involving ulnar osteotomy, open reduction, and annular ligament reconstruction, is necessary; however, the outcomes are generally inferior to those achieved with early intervention [2,9].

Despite their clinical relevance, Monteggia fracture-dislocations are frequently misdiagnosed in acute settings. The reported misdiagnosis rates range from 20% to 50% [2,10], particularly when the ulna shows plastic deformation without a visible fracture line, and the dislocated radial head is mistaken for a sprain [11]. In emergency departments, the focus is often on apparent ulnar injuries, and a proper assessment of the elbow joint can be neglected.

This study aimed to assess the diagnostic accuracy of Monteggia fractures among physicians of varying specialties and training levels, including radiologists, pediatric orthopedic surgeons, general orthopedic surgeons, and orthopedic residents. By identifying physician- and system-related factors associated with diagnostic performance, we hope to inform targeted educational efforts and systemic improvements that enhance early recognition and reduce missed or delayed diagnoses in pediatric patients.

## 2. Materials and Methods

### 2.1. Data Source

This retrospective study analyzed pediatric patients who underwent radiographic examinations for suspected elbow or forearm fractures at a tertiary referral hospital between 1 March 2010 and 31 December 2022. Eligible patients were identified through a comprehensive review of the hospital’s picture archiving and communication systems (PACS). A total of 1022 cases were initially reviewed by a senior pediatric orthopedic surgeon, 40 of which were identified as suspected Monteggia fractures. The cases were subsequently reviewed by an initial reviewer and an independent senior pediatric orthopedic surgeon. The diagnosis of Monteggia fracture was confirmed by consensus based on independent assessments of both initial and follow-up radiographs.

From this cohort, 120 cases were selected for diagnostic accuracy analysis. This sample included 40 confirmed Monteggia fractures (33.3%) and 80 randomly selected non-Monteggia fractures (66.7%), the latter of which served as controls. Data on patient demographics (sex and age) and clinical variables, including splint application and radiographic adequacy, were obtained from electronic medical records. Only patients with complete data for all the study variables were included in the final analysis. Patients were excluded if only initial radiographs were available, without follow-up imaging to confirm fracture union.

For each case, the participating physicians reviewed radiographs of the initial upper-extremity images obtained in relation to the trauma. These included images of the humerus, elbow, forearm, and wrist in either the unilateral or bilateral views. Radiographs acquired at an external institution and which included an official radiology report were used. Otherwise, the first set of radiographs obtained at our institution on the day of presentation was selected. Radiographs were organized into 120 individually numbered folders, with one folder for each patient. Participants were instructed to make a diagnosis based solely on the radiographs in each folder without access to other clinical information, such as patient history, mechanism of injury, or physical examination findings. No time limit was imposed for the image reviews and the participants were not informed of the specific study hypotheses or the fracture types being assessed. Any recognition of Monteggia fractures was based on clinical experience or independent suspicion during image interpretation. This approach was intended to replicate diagnostic conditions commonly encountered in emergency or outpatient clinical settings.

Radiographic adequacy was assessed to ensure consistent diagnostic quality. All cases were reviewed regardless of image quality, and each was classified as either adequate or inadequate based on standardized criteria. For the elbow lateral view, adequacy required superimposition of the distal humeral condyles, a clearly visible ulnohumeral joint space, and the presence of an “hourglass” or “figure-of-eight” configuration, indicating a true lateral projection. Lateral views of the forearm were considered adequate when the distal radius and ulna were superimposed, whereas anteroposterior (AP) views were considered adequate when there was minimal overlap of the distal radius and ulna, indicating a fully supinated position. The physicians were categorized into six groups, including one pediatric radiologist, one junior pediatric orthopedic surgeon (trained externally), one junior general orthopedic surgeon, four orthopedic residents (one from each year of training), and general radiologists (official radiological reports generated by various radiologists during routine clinical care). A general orthopedic surgeon and the orthopedic residents were trained under the supervision of a senior pediatric orthopedic surgeon at our institution.

### 2.2. Variable Definitions

The primary outcome variable was radiological misinterpretation, which was defined as any discrepancy between the initial interpretation provided by the evaluating physician and the final consensus diagnosis established by an expert pediatric orthopedist. Interpretations that completely matched the reference diagnosis were classified as correct, whereas deviations were classified as incorrect. Accordingly, the outcome variables were treated as binary (i.e., correct or incorrect).

The independent variables were classified as physician- and patient-related factors. Physician-related variables included the evaluator’s group, categorized into six types: pediatric radiologists, pediatric orthopedic surgeons, general orthopedic surgeons, orthopedic residents in each of the four years of training, and general radiologists (official radiology reports). Patient-related variables included sex (male or female), age group (<6 years vs. ≥6 years), radiographic adequacy (adequate vs. inadequate), and splint application status (presence or absence of a splint at the time of imaging). These variables were selected based on their clinical relevance and the existing evidence of their impact on the accuracy of fracture interpretation. An age cutoff of 6 years was chosen, as this is typically when the capitellum, radial head, and medial epicondyle are visible on elbow radiographs, whereas the trochlea, olecranon, and lateral epicondyle may remain unossified, potentially leading to misinterpretation [12].

### 2.3. Statistical Analysis

Descriptive statistics were used to summarize the baseline characteristics of the study population stratified by fracture type (Monteggia vs. other). The radiological interpretation accuracy and error rates were calculated for each physician group and fracture type. Univariate logistic regression analyses were performed to estimate the odds ratios (ORs) and 95% confidence intervals (CIs) for misinterpretation according to the physician type, with pediatric radiologists serving as the reference group. In addition, multivariate logistic regression analysis was performed to adjust for potential confounding factors. Covariates included sex, age, radiographic adequacy, and splint application. Separate analyses were conducted for all fractures, Monteggia fractures, and other fractures to examine the fracture-type-specific factors associated with misinterpretation. All statistical analyses were performed using the SAS software (version 9.4; SAS Institute Inc., Cary, NC, USA). Statistical significance was set at *p* < 0.05. No imputation for missing data was performed because only complete cases were included in the analysis.

### 2.4. Ethical Considerations

This study was approved by the Institutional Review Board (IRB) of the Pusan National University Yangsan Hospital Institutional Review Board (IRB no. 55-2025-022). The requirement for informed consent was waived due to the retrospective nature of the study and the minimal risk posed to the participants. All study procedures were conducted in accordance with the ethical standards of the institutional and/or national research committee and the 1964 Declaration of Helsinki and its later amendments, or comparable ethical standards.

## 3. Results

The characteristics of the study population according to fracture type (Monteggia vs. other) are summarized in Table 1. Of the 120 patients, 40 (33.3%) had Monteggia fractures, while 80 (66.7%) had other fracture types. Overall, the study cohort was predominantly male (59.2% male vs. 40.8% female patients). However, the proportion of female patients was higher in the Monteggia fracture group (52.5%) compared to the other fracture groups (35.0%).

Approximately half of the patients (48.3%) were under 6 years of age, with a similar distribution between the fracture types. Regarding patient positioning for imaging, 45.8% of all imaging studies were considered adequate for diagnostic purposes, while 54.2% were classified as inadequate. Adequate patient positioning was slightly more common in the Monteggia fracture group (52.5%) than in other fracture groups (42.5%). Regarding splint application, 30.8% of patients had a splint, with a comparable distribution between the two fracture groups.

The radiological interpretation error rates according to physician and fracture type (Monteggia vs. other) are shown in Table 2. Among the physician groups, first-year orthopedic residents exhibited the highest overall error rate (46.7%), followed by general radiologists (40.0%). In contrast, pediatric radiologists demonstrated the lowest error rate (12.5%).

In terms of fracture type, the error rates were consistently higher for Monteggia fractures than for other fracture types across all physician groups. Notably, the error rate for Monteggia fractures among first-year orthopedic residents was 87.5% compared to 26.3% for other fractures. Similarly, general radiologists reported a markedly higher error rate for Monteggia fractures (77.5%) compared to the other types of fractures (21.3%).

Overall, less-experienced physicians and those with less-specialized training had significantly higher error rates, particularly when interpreting Monteggia fractures.

Table 3 presents the radiological misinterpretation rates according to physician type compared with pediatric radiologists stratified by fracture type. Overall, general radiologists and orthopedic residents demonstrated significantly higher odds of misinterpretation than the pediatric radiologists (reference group).

For total fractures, the odds of misinterpretation were highest among first-year orthopedic residents (OR = 6.13, 95% CI: 3.20–11.72, *p* < 0.001), followed by general radiologists (OR = 4.67, 95% CI: 2.43–8.96, *p* < 0.001).

When stratified by fracture type, the odds of misinterpretation were markedly higher for Monteggia fractures than for other fracture types across all physician groups. First-year orthopedic residents had the highest odds of misinterpretation for Monteggia fractures (OR = 17.50, 95% CI: 5.88–52.10, *p* < 0.001), followed by general radiologists (OR = 13.78, 95% CI: 4.71–40.28, *p* < 0.001).

In contrast, for other fracture types, although the ORs were generally elevated among residents and general radiologists compared to pediatric radiologists, statistical significance was observed only in first-year residents (OR = 3.71, 95% CI: 1.48–9.33, *p* = 0.004) and general radiologists (OR = 2.81, 95% CI: 1.10–7.22, *p* = 0.027).

These findings suggest that both physician specialization and clinical experience significantly influenced the accuracy of radiological interpretation, particularly in challenging cases, such as Monteggia fractures.

Table 4 presents the adjusted ORs for radiological misinterpretation, focusing on the most representative physician groups from Table 3, namely pediatric radiologists, pediatric orthopedists, and first-year orthopedic residents. Other orthopedic resident groups (second-, third-, and fourth-year residents) demonstrated misinterpretation patterns similar to those of first-year residents and were therefore not separately presented.

In pediatric radiologists, splint application was significantly associated with increased odds of misinterpretation for total fractures (OR = 5.26; 95% CI: 1.55–17.80; *p* = 0.008) and non-Monteggia fractures (OR = 6.38; 95% CI: 1.01–40.51; *p* = 0.049), but not for Monteggia fractures.

In pediatric orthopedists, splint use was significantly associated with higher odds of misinterpretation across all fracture types: total fractures (OR = 5.63; 95% CI: 2.07–15.34; *p* < 0.001), Monteggia fractures (OR = 5.18; 95% CI: 1.01–26.60; *p* = 0.049), and non-Monteggia fractures (OR = 8.23; 95% CI: 1.58–42.85; *p* = 0.012). In addition, the odds of misinterpretation for Monteggia fractures were significantly lower when evaluating patients under 6 years of age (OR = 0.21; 95% CI: 0.04–0.99; *p* = 0.049), suggesting improved diagnostic accuracy in younger children within this group.

In contrast, none of the evaluated factors were significantly associated with misinterpretation in the assessments by the first-year orthopedic residents, regardless of fracture type.

## 4. Discussion

In this study, the diagnostic accuracy of pediatric Monteggia fractures was found to vary significantly based on physician experience, with a more evident disparity observed for Monteggia fractures than for other types of fractures. Junior clinicians, particularly first-year residents, had notably lower recognition rates than senior physicians, whereas experienced orthopedic surgeons and subspecialized radiologists performed better. These findings are in line with previous studies in which initial missed diagnosis rates of 20–30% were reported [2,13,14]. In a retrospective review of 220 pediatric forearm fractures, Gleeson et al. found that 50% of Monteggia fractures were missed by junior emergency physicians and 25% by senior radiologists. Even subspecialists have been reported to miss 20–28% of cases [10,15,16], highlighting that while diagnostic performance improves with experience, errors still occur across all levels.

Several factors may explain the frequent misdiagnosis of acute pediatric Monteggia fractures [10,17,18,19,20,21]. The sequential and age-dependent appearance of elbow ossification centers can obscure the radiographic assessment of radiocapitellar alignment, particularly when the radial head is not fully ossified. Therefore, subtle dislocations may not be recognizable. Furthermore, pediatric patients are frequently uncooperative during physical examinations and radiographic imaging, particularly in emergency settings, which can result in incomplete or suboptimal imaging findings. In addition, clinicians may focus excessively on the management of ulnar fractures while failing to assess the radiocapitellar joint, leading to missed diagnoses of Monteggia lesions. Cognitive biases, such as satisfaction with a search, may further contribute to diagnostic oversight. The phenomenon known as “satisfaction of search” (SOS) is a well-documented cognitive bias in radiology. This occurs when the detection of one abnormality leads to the premature termination of the search, resulting in additional abnormalities being overlooked. This bias is particularly pertinent in cases of Monteggia fractures, in which an obvious ulnar fracture may distract from a concurrent radial head dislocation [22,23]. These findings underscore the importance of system-level safeguards to complement clinical awareness and experience.

Monteggia fracture-dislocations are rare and often subtle on imaging, particularly when ulnar fractures are incomplete. Greenstick fractures or plastic deformation with associated radial head dislocation may go unnoticed unless specifically sought [2,10,11]. Notably, experienced physicians occasionally fail to diagnose Monteggia fractures. The complexity of the injury and the potential for human error may contribute to missed diagnoses even among skilled clinicians [10]. In our study, several cases were missed by the attending physicians, consistent with previous reports, indicating that no group was entirely immune to diagnostic errors.

Given their rarity and subtle radiographic presentation, the prompt recognition of Monteggia fractures is critical to avoid long-term elbow dysfunction. A particularly noteworthy deficit in our cohort was observed among first-year residents and radiologists without musculoskeletal specialization. Less experienced observers had a substantially lower accuracy in identifying Monteggia injuries. In fact, a lack of specialized expertise corresponded to higher missed diagnosis rates in initial radiographic interpretations. In our study, initial radiology reports by on-call general radiologists frequently failed to detect Monteggia injuries. Thus, although experience and training significantly improve detection, awareness remains paramount.

Awareness and structured training are essential for improving the recognition of Monteggia fractures. Our findings suggest that diagnostic accuracy may improve when clinicians maintain an active awareness of the possibility of a Monteggia injury. In our study, participants who considered this diagnosis to be informed by their prior training or clinical experience were more likely to correctly identify the injury. This supports the notion that a high index of suspicion plays a critical role in recognizing Monteggia fractures [24]. Herein, first-year residents showed the lowest diagnostic accuracy across all physician groups, whereas subspecialty-trained pediatric orthopedists and radiologists performed significantly better. This supports the notion that clinical experience and focused musculoskeletal training are the key determinants of diagnostic success in the identification of Monteggia fractures. Many missed cases were initially misdiagnosed as isolated ulnar fractures, with radial head dislocation recognized only later [25]. Therefore, clinicians should always consider this injury pattern in children presenting with ulnar fractures or post-traumatic elbow pain. Best practices include obtaining dedicated elbow radiographs and carefully assessing the radiocapitellar alignment in all views [26]. Accordingly, a thorough clinical and radiological assessment should be performed for pediatric forearm injuries, wherein comparison with the contralateral limb may aid in identifying subtle abnormalities [27,28].

Historically, diagnostic errors have been linked to soft tissue swelling, overlapping bones, or omission of the elbow joint on initial radiographs [21]. Previous studies have emphasized that focused attention on radiocapitellar alignment and subtle ulnar deformities can significantly improve detection [26]. Radiographic factors, such as image quality, splint presence, and patient age, had a minimal impact on the diagnosis of Monteggia in our study. Missed cases were primarily due to recognition failure rather than inadequate imaging. Nevertheless, missed diagnoses of Monteggia still occurred, indicating that cognitive errors were key factors. In contrast, for other fracture types, suboptimal image quality or splinting were more likely to have contributed to missed diagnoses. In the present cohort, most radiographs included proper elbow views, thus minimizing the technical limitations. Ultimately, an accurate diagnosis of Monteggia fractures depends more on careful inspection and familiarity than on image quality alone.

Our study also highlighted the high misdiagnosis rates of Monteggia fractures in routine radiological reports, showing that radiologists, particularly in busy emergency settings, at times overlook these injuries. Previous research has reported that Monteggia lesions are frequently omitted in initial interpretations, with approximately 70% recognized by both emergency clinicians and radiologists [11]. These findings support a team-based approach: clinicians should not rely solely on radiology reports, and radiologists should systematically assess radial head alignment. Safety measures, such as double reading or early specialist consultation, may reduce missed diagnoses, especially in institutions lacking on-site pediatric orthopedic expertise.

A multifaceted strategy is warranted to reduce the misdiagnosis of pediatric Monteggia fractures. Key components include improving early clinical training with structured instruction on radiographic assessment and common diagnostic pitfalls, as well as the use of simulations. Senior oversight is also crucial. Prompt review by attending physicians or radiologists can help to identify subtle dislocations missed by junior trainees. In cases of diagnostic uncertainty, short-interval follow-ups within 1–2 weeks may facilitate the detection of evolving radial head dislocations before a chronic deformity develops. Although our findings did not demonstrate a significant correlation between radiographic quality and diagnostic accuracy, previous studies have emphasized the importance of standardized imaging, including dedicated elbow views, for isolated ulnar fractures. Ensuring proper positioning and systematic evaluation of radiocapitellar alignment is essential, particularly in young children with incomplete ossification. Accordingly, a comprehensive clinical and radiological assessment, including contralateral comparison when needed, should be routinely performed for all pediatric forearm injuries. Collectively, these strategies highlight the importance of heightened clinical vigilance, structured training, and institutional safeguards to address the persistently high rate of missed Monteggia fractures in children. Early recognition depends not only on imaging but also on clinician awareness and suspicion. By integrating educational reinforcement, systematic image reviews, timely follow-up, and standardized radiographic protocols, healthcare providers can significantly reduce diagnostic errors and improve outcomes for this easily overlooked yet clinically significant injury.

In response to the growing interest in the application of artificial intelligence (AI) in musculoskeletal imaging, our findings suggest that AI could serve a complementary role in the diagnosis of pediatric Monteggia fractures. Accurate assessment of the radiocapitellar line—which should pass through the center of the capitellum on both AP and lateral views—is essential for detecting radial head dislocation, the hallmark of Monteggia injuries [29]. Proper radiographic positioning is also critical, as malrotated or misaligned images may obscure anatomical landmarks and compromise diagnostic accuracy.

AI-based tools trained to detect alignment abnormalities, such as radiocapitellar line deviation, and to assess the adequacy of radiographic projections could function as effective real-time quality control systems. For instance, automated recognition of whether an image represents a true AP or lateral view could help to ensure that diagnostic landmarks are reliably captured. Rather than replacing clinical judgment, such systems may support clinicians—particularly junior physicians or those working in high-volume emergency settings—by flagging suboptimal images and prompting closer evaluation. Future research should focus on the development and validation of AI-assisted tools tailored to pediatric musculoskeletal imaging, with attention given to usability in acute care and educational environments.

This study had several notable strengths. The inclusion of multiple physician groups ranging from junior residents to subspecialty-trained musculoskeletal radiologists enabled a comprehensive assessment of diagnostic accuracy across different levels of training and specialization. Additionally, the consideration of contextual factors, such as splint application, patient positioning, and age-related ossification, added important clinical relevance to the interpretation of diagnostic challenges.

However, this study has some limitations. First, the single-center retrospective study design may limit its generalizability to other clinical settings. Second, the number of cases was relatively small, and the number of physician groups was limited, which may have affected the robustness of the subgroup comparisons. Third, the determination of diagnostic accuracy relied on radiographic interpretation alone without correlation with clinical outcomes. Fourth, although radiographic adequacy was generally high in this cohort, variations in image quality and standardization may have influenced the results. Finally, we did not formally assess interobserver variability, which may have affected the consistency of interpretations among different types of physicians.

## 5. Conclusions

Monteggia fracture-dislocations in children are often missed, especially by less experienced clinicians, owing to subtle radiographic signs and cognitive bias. Although diagnostic accuracy improves with experience, errors can occur at any level. The key strategies include structured education, senior review, standardized imaging, and early follow-up of uncertain cases. Addressing both individual and systemic factors can improve the early recognition and outcomes of this often-overlooked injury.

## Figures and Tables

**Figure 1 medicina-61-01457-f001:**
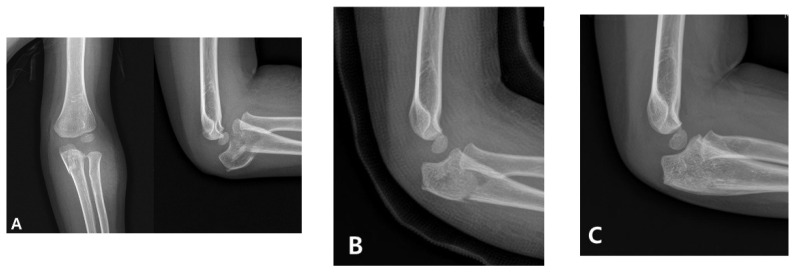
Radiographic progression of an acute pediatric Monteggia fracture treated conservatively. (**A**) Initial anteroposterior and lateral elbow radiographs of a 3-year-old boy revealing anterior dislocation of the radial head with a fracture of the proximal ulna, consistent with a Bado type I Monteggia fracture. (**B**) Closed reduction and casting were performed promptly. Post-reduction images confirmed the anatomical realignment of both the ulna and radial head. (**C**) Follow-up radiographs obtained 2 months after the injury demonstrate maintained reduction and progressive healing without the need for surgical intervention.

**Figure 2 medicina-61-01457-f002:**
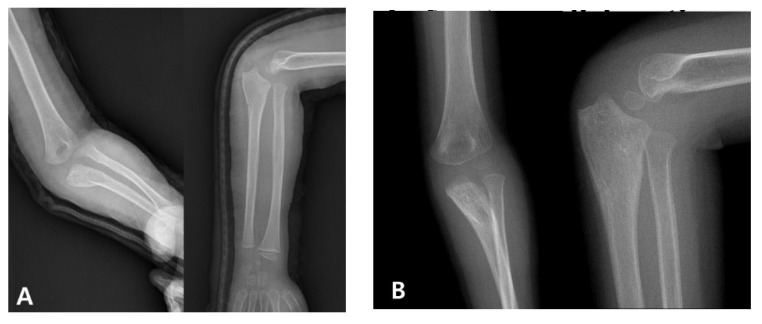
Radiographic presentation of a neglected pediatric Monteggia fracture. (**A**) Initial radiographs of a 4-year-old boy demonstrating a proximal ulnar fracture with anterior radial head dislocation. The injury was initially misdiagnosed as an isolated ulnar fracture, and the patient received conservative treatment at another institution. (**B**) Follow-up images taken after 6 weeks show fracture healing but persistent dislocation of the radial head, consistent with a neglected Monteggia fracture.

**Figure 3 medicina-61-01457-f003:**
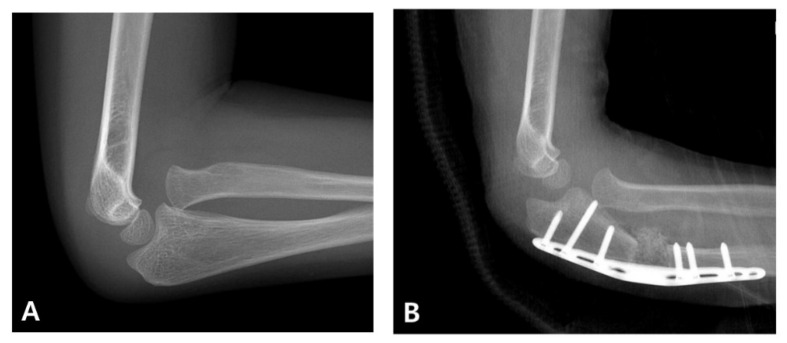
Surgical correction of a neglected Monteggia fracture-dislocation. (**A**) Preoperative radiographs of a 5-year-old boy showing persistent anterior dislocation of the radial head due to a previously unrecognized Monteggia lesion. (**B**) The patient underwent corrective ulnar osteotomy and open reduction surgery. Postoperative imaging confirmed the successful reduction and restoration of the radiocapitellar alignment.

**Table 1 medicina-61-01457-t001:** General characteristics of the study population according to Monteggia and other fracture types.

		Total Fracture		Monteggia Fracture		Other Fracture	
		N	%	N	%	N	%
Total		120	100.00	40	100.00	80	100.00
Sex	Female	49	40.83	21	52.50	28	35.00
	Male	71	59.17	19	47.50	52	65.00
Age group (Years)	Under 6	58	48.33	21	52.50	37	46.25
	6 and over	62	51.67	19	47.50	43	53.75
Radiographic adequacy	Adequate	55	45.83	21	52.50	34	42.50
	Inadequate	65	54.17	19	47.50	46	57.50
Splint	No	83	69.17	27	67.50	56	70.00
	Yes	37	30.83	13	32.50	24	30.00

**Table 2 medicina-61-01457-t002:** Radiological interpretation accuracy and error rates according to type of physician and fracture ^1^.

			Total Fracture		Monteggia Fracture		Other Fracture	
			N	%	N	%	N	%
Total			120	100.00	40	100.00	80	100.00
Pediatric Radiologist		Correct	105	87.50	32	80.00	73	91.25
		Incorrect	15	12.50	8	20.00	7	8.75
Pediatric Orthopedist		Correct	95	79.17	25	62.50	70	87.50
		Incorrect	25	20.83	15	37.50	10	12.50
General Orthopedist		Correct	95	79.17	24	60.00	71	88.75
		Incorrect	25	20.83	16	40.00	9	11.25
Orthopedic Residents	Fourth-year	Correct	85	70.83	21	52.50	72	90.00
		Incorrect	35	29.17	19	47.50	8	10.00
	Third-year	Correct	85	70.83	18	45.00	67	83.75
		Incorrect	35	29.17	22	55.00	13	16.25
	Second-year	Correct	80	66.67	14	35.00	66	82.50
		Incorrect	40	33.33	26	65.00	14	17.50
	First-year	Correct	64	53.33	5	12.50	59	73.75
		Incorrect	56	46.67	35	87.50	21	26.25
General Radiologists		Correct	72	60.00	9	22.50	63	78.75
		Incorrect	48	40.00	31	77.50	17	21.25

^1^ Accuracy was defined as the proportion of correct interpretations compared to the reference standard. An error was defined as any misinterpretation leading to an incorrect diagnosis.

**Table 3 medicina-61-01457-t003:** Rates of radiological misinterpretation according to physician type compared with pediatric radiologists, stratified by fracture type.

		Total Fracture				Monteggia Fracture				Other Fracture			
		OR	95% CI		*p*	OR	95% CI		*p*	OR	95% CI		*p*
			LL	UL			LL	UL			LL	UL	
Pediatric Radiologist		Reference				Reference				Reference			
Pediatric Orthopedist		1.84	0.92	3.70	0.083	2.40	0.88	6.56	0.084	1.49	0.54	4.13	0.442
General Orthopedist		1.84	0.92	3.70	0.083	2.67	0.98	7.25	0.051	1.32	0.47	3.74	0.598
Orthopedic Residents	Fourth-year	2.88	1.48	5.63	0.002	3.62	1.34	9.77	0.009	1.16	0.40	3.36	0.786
	Third-year	2.88	1.48	5.63	0.002	4.89	1.81	13.21	0.001	2.02	0.76	5.37	0.152
	Second-year	3.50	1.81	6.78	<0.001	7.43	2.70	20.42	<0.001	2.21	0.84	5.81	0.101
	First-year	6.13	3.20	11.72	<0.001	17.50	5.88	52.10	<0.001	3.71	1.48	9.33	0.004
General Radiologists		4.67	2.43	8.96	<0.001	13.78	4.71	40.28	<0.001	2.81	1.10	7.22	0.027

OR, odds ratio; CI, confidence interval; LL, lower limit; UL, upper limit.

**Table 4 medicina-61-01457-t004:** Adjusted ORs for misinterpretation according to physician and fracture type using multivariate logistic regression analysis.

		Total Fracture				Monteggia Fracture				Other Fracture			
		Adj OR ^1^	95% CI		*p*- Value	Adj OR ^1^	95% CI		*p*- Value	Adj OR ^1^	95% CI		*p*- Value
			LL	UL			LL	UL			LL	UL	
Pediatric Radiologist													
Sex	Female	2.37	0.74	7.63	0.147	3.51	0.54	22.94	0.189	1.77	0.31	10.04	0.518
Age group (Years)	Under 6	2.02	0.61	6.75	0.252	0.70	0.12	3.98	0.688	6.64	0.71	62.30	0.098
Radiographic adequacy	Inadequate	0.91	0.27	3.06	0.876	1.79	0.29	10.95	0.527	0.66	0.11	3.88	0.646
Splint	Yes	5.26	1.55	17.80	0.008	4.16	0.70	24.61	0.116	6.38	1.01	40.51	0.049
Pediatric Orthopedist													
Sex	Female	1.05	0.41	2.75	0.915	1.37	0.31	6.10	0.677	0.32	0.05	1.89	0.207
Age group (Years)	Under 6	0.46	0.17	1.23	0.121	0.21	0.04	0.99	0.049	0.61	0.12	3.03	0.549
Radiographic adequacy	Inadequate	1.23	0.46	3.30	0.687	1.34	0.28	6.51	0.717	2.61	0.47	14.51	0.273
Splint	Yes	5.63	2.07	15.34	<0.001	5.18	1.01	26.60	0.049	8.23	1.58	42.85	0.012
First-year Resident													
Sex	Female	1.50	0.71	3.13	0.285	2.07	0.24	17.90	0.507	0.89	0.31	2.60	0.837
Age group (Years)	Under 6	0.94	0.45	1.97	0.873	1.87	0.20	17.74	0.585	0.57	0.20	1.64	0.295
Radiographic adequacy	Inadequate	0.76	0.36	1.61	0.476	2.46	0.21	29.28	0.477	0.68	0.24	1.91	0.465
Splint	Yes	1.61	0.71	3.65	0.251	1.80	0.27	4.91	0.942	1.53	0.49	4.76	0.463

^1^ Adj OR: odds ratio adjusted for sex, age group, radiographic adequacy, and splint use. Reference groups are male (for sex), age 6 years and over (for age group), adequate imaging, and no splint. Abbreviations: OR, adjusted odds ratio; CI, confidence interval; LL, lower limit; UL, upper limit.

## Data Availability

Data is contained within the article.

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
