# Peer review of "Challenges in Diagnosing Pediatric Monteggia Fractures: Role of Clinical Experience and Imaging"

_medicina, 2025, doi:10.3390/medicina61081457_

Round 1
Reviewer 1 Report
Comments and Suggestions for Authors
This is a well-written and relevant study focusing on a well-known clinical challenge: the missed diagnosis of Monteggia fractures in children. The study aimed to evaluate the diagnostic accuracy among physicians of varying specialties and training levels.
Here provide some recommendations
Methodology: Although the cases were collected retrospectively from clinical works, the test to various level of expertise would be in a format of prospective recruitment and protocols, for example, design of questionnaire, time allowed for answer, sub-classification of Monteggia etc. However, this part had not been addressed clearly. Besides, the definition of “inadequate positioning” was not clearly addressed.
Results: the statements from the results were appropriate and compatible with statistics analysis.
Discussion: The discussion effectively interprets the findings, relates them to existing literature, and offers practical recommendations for improving diagnostic accuracy. However, most of the suggestions are based on literatures review.
Between line 303-306, “our findings suggest that simply knowing to look for …….”, means some intervention had been applied during this study. Could you address it more clearly on Materials and Methods and provide data on Results.
General recommendation
Specificity on "System-Level Interventions": The title mentions "System-Level Interventions," but the manuscript could benefit from a more explicit discussion and examples of what these interventions entail beyond education and standardized protocols.
I recommend this manuscript for publication after the above issues being well addressed. In spite of limitations from the study design, the Discussion provides valuable insights into the challenges of diagnosing pediatric Monteggia fractures and offers practical recommendations for improving diagnostic accuracy. I hope these comments are helpful
Reviewer 2 Report
Comments and Suggestions for Authors
The authors present a retrospective study comparing the diagnostic accuracy of different physician groups in evaluating Monteggia fractures in pediatric patients. Overall, the study is well conducted and clearly written.
However, there are areas that could enhance its readability. In particular, I recommend reducing the length of the introduction and discussion sections, as the latter tends to be repetitive.
A few specific points:
- How many physicians were included in each group?
- In the current era of AI, it would be interesting to discuss its potential applications in assisting the diagnosis of these challenging fractures. Please streamline the discussion section to reduce repetitive content and incorporate a section on future directions.
In conclusion, while the manuscript is of good quality, some revisions are necessary to improve its clarity and readability.
Round 2
Reviewer 2 Report
Comments and Suggestions for Authors
The authors have adequately addressed my requests. The manuscript is now suitable for publication from my perspective.
Author Response
Comment: The authors have adequately addressed my requests. The manuscript is now suitable for publication from my perspective.
Response : We sincerely thank the reviewer for the positive feedback and for recognizing the revisions made to improve the manuscript. We greatly appreciate your time and effort in reviewing our work.